# Metformin Therapy and Breast Cancer Incidence in the Ha’il Region

**DOI:** 10.3390/healthcare11030321

**Published:** 2023-01-20

**Authors:** Mhdia Osman, Taif Muqbel, Ahad Abduallh, Shuruq Alanazi, Nasrin E. Khalifa, Weam M. A. Khojali, Halima Mustafa Elagib, Weiam Hussein, Marwa H. Abdallah

**Affiliations:** 1Department of Clinical Pharmacy, Faculty of Pharmacy, University of Ha’il, P.O. Box 2440, Ha’il 55476, Saudi Arabia; 2Student College of Pharmacy, University of Ha’il, Ha’il 55476, Saudi Arabia; 3Department of Pharmaceutics, Faculty of Pharmacy, University of Ha’il, Ha’il 55476, Saudi Arabia; 4Department of Pharmaceutics, Faculty of Pharmacy, University of Khartoum, Khartoum 13314, Sudan; 5Department of Pharmaceutical Chemistry, Faculty of Pharmacy, University of Ha’il, Ha’il 55476, Saudi Arabia; 6Department of Pharmaceutical Chemistry, Faculty of Pharmacy, Omdurman Islamic University, Khartoum 14415, Sudan; 7Department of Pharmacology, Faculty of Medicine, University of Ha’il, Ha’il 55476, Saudi Arabia; 8Department of Pharmacology, College of Pharmacy, Omdurman Islamic University, Khartoum 14415, Sudan; 9Department of Pharmaceutical Chemistry, Faculty of Pharmacy, Aden University, Aden 6075, Yemen; 10Department of Pharmaceutics and Industrial Pharmacy, Faculty of Pharmacy, Zagazig University, Zagazig 44519, Egypt

**Keywords:** breast cancer, diabetes, metformin

## Abstract

Background: Metformin is a drug used to treat patients with type 2 diabetes, especially those who suffer from obesity. It is also used in the treatment of women with polycystic ovary syndrome (PCOS). This disease is related to insulin resistance and multiplied blood sugar ranges. Furthermore, it has been established that the use of metformin improves the menstrual cycles and ovulation rates of these women. Methods: A structured questionnaire was conducted to determine the prevalence of breast cancer among women using metformin in the Ha’il region. Result: The incidence of breast cancer among women using metformin in the Ha’il region is very low. Thus, it can be said that breast cancer cases declined among diabetics taking metformin. This means that metformin use is associated with a lower risk of breast cancer in women with type 2 diabetes, even in cases where these women have a family history of breast cancer. Conclusions: According to previous findings, metformin has been linked to lower breast cancer risk in women with type 2 diabetes. Furthermore, the findings of this study corroborate the literature on this subject by indicating that there is a substantial connection between metformin use and a lower risk of breast cancer in women with type 2 diabetes. However, further in vitro and in vivo experiments are crucial to investigate the protective effect of metformin against breast cancer and to confirm our findings.

## 1. Introduction

Diabetes treatments such as metformin affect the risk and the incidence of breast cancer [1]. Metformin is an oral, hypoglycemic, biguanide used mainly to treat type 2 diabetes mellitus (T2D). Evidence shows that metformin is associated with increased cardiovascular mortality [2] and reduced risk of certain cancers (e.g., breast cancer) in addition to improving glycemic regulation [1].

Breast cancer is a heterogeneous disease that affects the lymph nodes in the armpits and other organs, starting as a local lesion in the breast and then spreading progressively [3]. Several factors serve as alarms and decide the type of cancer, treatment, and outcome [4]. In a study conducted in the United Kingdom, they found that most ladies with type 2 diabetes who have used metformin for many years showed reduced chances of breast cancer, whilst no such impact was found in cases of temporary use. When they excluded insulin users from the assessment, the correlation was close [5].

A systematic review carried out in Kazakhstan discussed the possible anticancer mechanism of metformin and the significance of AMP-activated protein kinase (AMPK) as an expected target for anticancer treatments [6].

According to some articles from the *Journal of Clinical Oncology of the American Society of Clinical Oncology*, the incidence of breast cancer in women with diabetes who were not taking metformin treatment was significantly higher than the incidence of breast cancer in women with diabetes who were receiving metformin [7]. Furthermore, a review article discussed multiple molecular properties of metformin, such as reactive oxygen species inhibition; inhibition of mTORC1, thus acting as an antitumor agent; modulation of ADORA; and the activation of AMPK, all of which point to the benefits of metformin [8]. Additionally, a study discussed the use of different doses of metformin, i.e., 5, 10, 20, 50, and 100 mM, to challenge breast cancer cells (MCF-7) and colon cancer cells (CaCo-2). A substantial reduction in the cell count of both colon and breast cancer cells was revealed using the Trypan blue assay [9].

In a study focused on determining whether metformin use can be linked to a change in the pathologic complete response (PCR) rates in diabetic patients with breast cancer after neoadjuvant chemotherapy, it was discovered that the diabetic patients using metformin showed a greater PCR rate than diabetics who did not take metformin for their diabetes [10]. For the first time, metformin’s potential antitumor characteristics were revealed in 2005 by Scottish researchers, who discovered that T2D patients using the drug appeared to have a decreased cancer risk [11]. Metformin is believed to have a positive effect on immune responses against tumor cells and metabolic pathways within cells, mainly through the activation of AMPK (adenosine monophosphate-activated protein kinase) (Figure 1) [12].

Moreover, another report, which studied the ability of metformin to prevent the proliferation of breast cancer cells, found that when used in vitro, it decreases colony formation. For breast cancer patients, ER and erbB2 are important prognostic markers and therapeutic targets. In the said study, four breast cancer cell lines with variable levels of the expression of these two markers were used to research the effects of metformin used in vitro, including MCF-7 and MCF-7/713 (MCF-7 transfected with erbB2, BT-474, and SKBR-3) [13].

Although there are many studies, statistics, and conclusions about the role of metformin in breast cancer, a review was conducted in various regions of Saudi Arabia, including the city of Ha’il, which showed that there is no relationship between breast cancer and diabetes. Thus, despite Ha’il having the highest percentage of diabetes mellitus, the results showed the lowest percentage of breast cancer in the city [14].

The aim of this study was to investigate the incidence and risk of breast cancer among diabetic patients using metformin in the Ha’il population.

## 2. Materials and Methods

A cross-sectional study was carried out among Ha’il women with type 2 diabetes using metformin with the help of direct patient interviewing in the Specialized Diabetes Center in Ha’il at King Salman, and through an online self-administrated questionnaire via Google forms. Women with diabetes having a family history of breast cancer were included. A convenient sample size (*n* = 257) was selected. Patient inquiries included information about patients’ demographic data, medication information, comorbid conditions, medication adherence, breast cancer diagnosis, family history of ovarian or breast cancer and relationship degree, and the type of management used for breast or ovarian cancer. The data gathered were analyzed using SPSS Version 23 and Statistical Package for Social Sciences, with endnote citations. In the results, *p* value was considered significant when *p* ≤ 0.05 [15]. Both Fisher’s exact test for categorical variables and the chi-square test (X_2_) were used to determine the statistical significance across the groups, and *p* values under 0.05 were regarded as statistically significant.

## 3. Results

### 3.1. Patients’ Demographic Data

About 38.9% of the patients were aged more than 55 years, with 42.8% having university-level education, and 30.7% of the patients were aged between 46 and 55 years, with 27.2% having primary education. About 27.2% of the patients were aged between 35 and 45 years, with 21.8% having secondary-level education, and the least number of the patients (3.1%) were aged between 25 and 35 years, with 8.2% having an intermediate education level (Table 1).

### 3.2. Medication Information

About 98.8% of the patients used metformin for the treatment of diabetes, whereas only 1.2% did not use metformin. Out of the patients using metformin, 38.6% used metformin for 5–10 years, 37% of them used metformin for less than 5 years, and 24.4% used metformin for more than 10 years. Notably, 68.1% of the patients did not take any additional medication with metformin, and 31.9% of the patients took medication in addition to metformin. The major medications used alongside metformin were insulin (45.1%), linagliptin (20.7%), sitagliptin (11%), glimepiride (8.5%), empagliflozin and sitagliptin (8.5%), gliclazide and sitagliptin (4.9%), and empagliflozin (1.2%) (Table 2).

### 3.3. Comorbidities

About 62.3% of the patients had comorbid diseases, and 37.7% did not have any other disease apart from type 2 diabetes militias (DM) (Figure 1).

### 3.4. The Type of Other Comorbid Conditions 

As shown in Figure 2, out of all the comorbid diseases that patients were suffering from, 50.5% of the patients suffered from hypertension, 24.7% from a thyroid disorder, 6.2% from hyperlipidemia, 4.1% from hypercholesterolemia, 3.1% from rheumatoid arthritis, 3.1% from hypertension with hypercholesterolemia, 2.1% from hypertension and thyroid, 2.1% from anemia, 2.1% from hypertension with depression, 1% from asthma, and 1% from ventricular septal defect.

### 3.5. Breast Cancer Diagnosis

The majority of diabetic patients (97.3%) did not have breast cancer, while only 2.7% were diagnosed with breast cancer (Figure 3).

### 3.6. Relationship Degree (for Participants Who Had a Family History of Breast or Ovarian Cancer)

First-degree relatives accounted for 7%, 4.7% were second-degree relatives, third-degree relatives accounted for 3.5%, and 1.6% were fourth-degree relatives (Figure 4).

### 3.7. The Relationship between Medication Group and Patient Diagnosis of Breast Cancer

The majority of the patients who took metformin as monotherapy (67.3%) did not suffer from breast cancer, and only 0.78%, i.e., 28.9% of all the patients, were diagnosed with breast cancer; this was followed by patients taking metformin and insulin, with 14.4% not suffering from breast cancer; however, the total parentage of patients using other medication with metformin had a higher percentage of breast cancer (1.92%, i.e., 71.1%) (Figure 5).

### 3.8. The Relationship between Comorbid Disease and Patient Diagnosis of Breast Cancer

The percentage of patients having no comorbid disease and diagnosed with breast cancer (1.56%, i.e., 57.14% of breast cancer patients) was more than the percentage of those having a comorbid disease (1.17%, i.e., 42.86% of breast cancer patients) (Figure 6). 

### 3.9. The Relationship between Medication Group and Patient Diagnosis of Breast Cancer

The percentage of patients diagnosed with breast cancer who had a family history of breast cancer (2.33%, i.e., 85.35% of breast cancer patients) was higher than the percentage of those with no family history (0.39%, i.e., 14.29% of breast cancer patients) (Figure 7).

## 4. Discussion

Breast cancer is the second leading cause of cancer-related deaths in women worldwide. More than 1,000,000 cases of breast cancer and more than 380,000 deaths from breast cancer were recorded in the year 2000 [16]. Metformin is an antidiabetic agent that reduces the risk of breast cancer [17]. 

A majority of diabetic patients (i.e., 38.95% aged >55 years with university education; 30.7% of the patients aged between 46 and 55 years with primary education; 27.2% of middle-aged patients, i.e., those aged between 35 and 45 years with secondary education; and 3.1% young adults, i.e., patients aged between 25 and 35 years with intermediate education) were older than 46 years of age. This finding is supported by another study, where it was found that obese females aged 41–60 years significantly have DM [18].

Almost all diabetic patients (98.8%) used metformin in their regimens. A majority (38.6%) of these patients used metformin for 5–10 years, followed by 37% who used it for less than 5 years, and 24.4% who used metformin for more than 10 years. Thus, metformin was noted to be a widely used drug among Ha’il citizens, and this is supported by Hundal Ripudaman et al. (2003), who concluded that metformin is the most widely prescribed antidiabetic agent [19].

The majority of these patients, i.e., 68.1%, used metformin as a monotherapy drug. Only 31.9% used metformin alongside other diabetic medication, such as insulin (45.1%), linagliptin (20.7%), sitagliptin (11%), glimepiride or empagliflozin and sitagliptin (8.5%), and a minimum number of patients (1.2%) used empagliflozin alone with metformin. The cytoprotective, oncostatic benefits of oral antidiabetic medications are supported by a meta-analysis, which also demonstrates that thiazolidinedione is connected to a lower risk of cancer, notably colorectal and breast cancers [20].

A majority of diabetic patients (62.3%) did not suffer from any comorbid disease along with diabetes, and only 37.7% suffered from other disease(s) along with diabetes. Out of these diseases, hypertension was found to be a predominant occurring disease (50.5%) among diabetic patients. Another study by Long and Amanda, who showed that approximately 75% of patients with diabetes have concomitant hypertension [21], also supports this finding. The second disease predominantly occurring among diabetic patients was thyroid disorder (26.7%), followed by hyperlipidemia (6.2%), hypercholesterolemia (4.1%), rheumatoid arthritis (3.1%), hypertension with hypercholesterolemia (3.1%), hypertension and thyroid (2.1%), anemia (2.1%), hypertension with depression (2.1%), asthma (1%), and ventricular septal defect (1%).

A majority of the patients (97.3%) did not suffer from breast cancer, and only 2.7% had breast cancer. Thus, it was concluded that breast cancer cases declined among diabetics taking metformin. This means that metformin use is associated with a lower risk of breast cancer in women with type 2 diabetes, even in cases where these women have a family history of breast cancer. This is supported by Arif et al. [22], who concluded that type 2 diabetic women on prolonged treatment with metformin have a low risk of breast cancer. This further indicated that a significant relationship exists between the use of metformin and its effect on decreasing the risk of breast cancer in women with type 2 diabetes. Moreover, our findings are in accordance with Gong et al., who concluded that metformin appears to decrease the risk of pancreatic cancer and improve survival in diabetics with pancreatic cancer [23]. In addition, Shi et al. suggested that metformin is associated with survival benefits in patients with pancreatic cancer and concurrent diabetes mellitus [24]. 

About 83.3% of the patients did not have a family history of breast cancer, while 16.7% of the patients had a family history of breast cancer with first-degree relatives (7%), followed by 4.7% of second-degree relatives, 3.5% of third-degree relatives, and a minimum number of fourth-degree relatives (1.6%), i.e., patients having high percentages of family history.

Patients who used only metformin (64.3%) or metformin with insulin (14.4%) were less likely to have breast cancer (i.e., metformin and metformin with insulin decrease the risk of breast cancer); no studies in the literature mention both medications (i.e., insulin and metformin).

Fewer patients with comorbid diseases were diagnosed with breast cancer (1.2%) than those without comorbid diseases (1.6%), which indicates that comorbid diseases do not increase the risk of breast cancer; this is supported by Wu et al. (2015), who revealed that the risk of breast-cancer-specific mortality was significantly increased among women with a history of diabetes or myocardial infarction [25].

The number of patients diagnosed with breast cancer who had a family history of breast cancer was higher (2.33%, i.e., 85.35% of breast cancer patients) than the number of those with no family history (0.39%, i.e., 14.29% of breast cancer patients). Therefore, family history increases the risk of breast cancer among diabetic patients using metformin medication. This is supported by Salinas-Martínez et al. (2014), who concluded that women with prediabetes and diabetes should be considered a more vulnerable population for early breast cancer detection [26]. Our research indicated that metformin may reduce the incidence of breast cancer in women with type 2 diabetes. Furthermore, the results of this study support the literature on the subject by showing that metformin use is significantly associated with a decreased risk of breast cancer in women with type 2 diabetes.

## 5. Conclusions

Metformin is the preferred medication for Ha’il region patients. Insulin with metformin is the most popular medication. The majority of the diseases in the Ha’il region are diabetes and hypertension. The incidence of breast cancer disease decreases among diabetic patients who use metformin treatment in the Ha’il region in Saudi Arabia. The majority of patients take their medication regularly, and this shows a clear relationship between good adherence and low incidence of breast cancer, although diabetes patients in the Ha’il region have a high percentage of family history of breast cancer. Comorbid diseases for diabetic patients taking metformin do not affect the risk of breast cancer. However, further in vitro and in vivo experiments are crucial to investigate the protective effect of metformin against breast cancer and to confirm our findings.

## Data Availability

Not applicable.

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
