# Peer review of "Metformin Therapy and Breast Cancer Incidence in the Ha’il Region"

_healthcare, 2023, doi:10.3390/healthcare11030321_

Round 1

Reviewer 1 Report (Previous Reviewer 2)

1. The authors are requested to incorporate the molecular mechanism(s) associated with Metformin mediated therapy for breast cancer.

2. The authors also requested to provide some evidence associated with the pancreatic cancer risk is reduced by metformin treatment in patients with diabetes.

3. The authors are also advised to provide some in vitro and in vivo experiments in support of their findings.

Author Response

Reviewer 2 Report (Previous Reviewer 1)

Authors have addressed the issues and manuscript should be accepted in the current form.

Author Response

Reviewer 3 Report (New Reviewer)

The current manuscript presents and interesting study on the relationship between breast cancer and metformin intake in the Ha´il region. Its findings are somewhat interesting, but the manuscript lacks a lot, such as:

- Careful review of written English should be done; for example in the abstract where it says “A structured questionnaire will be conducted” it should be “A structured questionnaire was conducted”, because it has already been done, and hence is past tense; or in line 175, where it says “The percentage of patient diagnose with breast cancer” and should say “The percentage of patient diagnosed with breast cancer”; the whole manuscript should be reviewed;

- In the abstract, a better bridge should be done between background and methods, including the research question; also the results should be better explained (not just 1 sentence); and conclusions should be better explained by said results, without excessive generalizations;

- More references should be added to the introduction section to better support what is being said;

- The sentence “Metformin is believed to have positive effect on immune responses against tumor cells and metabolic pathways within cells mainly through activation of AMPK (Adenosine Monophosphate–Activated Protein Kinase)(Figure 1) [12]” is misplaced, does not make sense to be located where it is;

- Also in the introduction section there is a bridge missing to the methods, stating what is really missing in the scientific literature about the subject that would justify the study;

- Sometimes the name of the city is written like “Hail” and others “Ha’il”, which is the correct one?

- In the results section, subsections are too short, sometimes only being one sentence and one figure;

- Quality of the images should be improved, especially figure 6, 7 and 8;

- The discussion section needs a final paragraph stating the general conclusions of the study;

- “We recommend in vivo animal model studies.” – which studies? To conclude what? And aren’t there studies about this already?

Round 2

Reviewer 3 Report (New Reviewer)

The manuscript has been improved.

This manuscript is a resubmission of an earlier submission. The following is a list of the peer review reports and author responses from that submission.

Round 1

Reviewer 1 Report

- I would suggest the authors to get the manuscript checked with a native English speaker or use editing services. There are lot of grammatical errors which makes it very difficult to understand the manuscript. 

- The font size from 67-70 is not same. Also have grammatical error. Correct it. 

- To make the manuscript more readable, authors can take effort to include a figure showing the MOA of Metformin and findings related to its effect cancer suppression. 

- Can authors discuss which category these 2.7% (Metformin consumers with breast cancer) patients belong to. Did they have first degree relatives? What were their comorbidities?

- I would request the authors to discuss their findings. Just presenting the results, is not very ideal for a manuscript to be considered for acceptance. 

- Line 148-49: Sentence is not complete or have errors. Correct it:- was noted to be a widely used drug among Hail citizens,this sborted by (-------)who said that; Metformin is the most widely prescribed antidiabetic agent(16)

- Nowhere in the manuscript its written that patients concern was taking with the certain ID and is included as attachment. 

Reviewer 2 Report

1. Authors are requested to provide possible molecular mechanism(s) in support of their study.

2. Authors are also requested to provide some in vivo animal model studies in support of their study.

Reviewer 3 Report

This manuscript has some serious problems.

- the layout is badly set up

- the English language is incomprehensible

- it is not clear what the endpoint of the study is, nor how you arrived at the conclusions described